# GRAPE: Let GPRO Supervise Query Rewriting by Ranking for Retrieval

## Abstract

The CLIP model has become a cornerstone of large-scale retrieval systems by aligning text and image data in a unified embedding space. Despite its simplicity and efficiency, CLIP struggles when applied to tasks whose input distributions diverge from its training corpus, such as queries with multilingual, long-form, or multimodal differences. To avoid costly retraining, existing methods mainly adopt query-rewriting strategies with large language models (LLMs), aiming to mitigate distribution gaps at the query level. However, due to the lack of supervision signals, LLMs fail to generate the optimal one that fits the training distribution. We address this challenge with GRAPE (**G**rouped **R**anking-**A**ware **P**olicy Optimization **E**nhancement), a plug-and-play enhancement approach that incorporates ranking signals into retrieval-guided query rewriting with LLMs. Intuitively, GRAPE proposes to leverage GRPO to bridge distributional differences—including length, multilingual, and modality shifts—by transforming queries into forms better aligned with the retriever's training distribution. However, our preliminary experiment finds that naively finetuning LLM with similarity scores can lead to score inflation, where nearly all candidates are assigned unexpectedly high scores regardless of their true relevance. To address score inflation, we propose a corpus-relative ranking-based reward, which explicitly aligns optimization with ranking metrics while suppressing spurious score inflation. Extensive experiments demonstrate that GRAPE consistently improves retrieval performance under distributional shifts—including multilingual differences (Flickr30k-CN, CVLUE, XM3600), length differences (Wikipedia), and multimodal differences (CIRR)—achieving an average improvement of 4.9% in Recall@10.

## 1 Introduction

The CLIP (Contrastive Language–Image Pretraining) model (Radford et al., 2021) learns a unified semantic space that aligns images and text. Owing to its simplicity and efficiency, tens-of-billions–scale vector indexes (Chuang et al.; Wang et al., 2025a) have been deployed in industry, with CLIP serving as a foundational encoder for downstream retrieval tasks (Stevens et al., 2024), as well as related applications such as zero-shot classification (Qu et al., 2025b; Martin et al., 2024; Wu et al., 2025) and clustering (Islam et al.; Qu et al., 2025a; Lowe et al., 2023). However, as retrieval-oriented downstream tasks continue to expand, tasks whose input distributions diverge substantially from CLIP's training corpus distribution often exhibit degraded performance.

The most straightforward approach is to expand the training corpus distribution to cover the distribution of downstream tasks (Fan et al., 2023; Huang et al., 2024; Yuksekgonul et al., 2022; Chen et al., 2022; Zhang et al., 2024; Wu et al., 2024; Sam et al., 2024), and then finetune the model on domain-specific datasets to yield notable performance improvements. However, a clear side effect of retraining is that it alters the learned embedding distribution, forcing costly re-embedding of all existing data and redeployment of downstream applications. Given the massive scale of vectorized data, such re-embedding is prohibitively expensive and disruptive. Consequently, how to enhance existing CLIP models without redeployment has become a critical and practical challenge.

A promising direction is to keep the retrieval pipeline unchanged while mitigating distribution gaps from the query side by leveraging large language models (LLMs) for query rewriting. In particular,

these methods typically design prompts that guide LLMs to rewrite downstream queries into forms that better match the retriever's original training distribution, thereby improving retrieval performance. However, such direct rewriting is often ineffective because LLMs lack awareness of CLIP's training distribution and thus cannot consistently generate high-quality rewrites. As an alternative, some subsequent works attempt to bypass this alignment issue—for example, by decomposing queries and images into sub-parts before retrieval to achieve finer-grained alignment (Jiang et al., 2022), or by incorporating feedback from LLM–user interaction to enrich queries and improve performance (Lee et al., 2024). Yet these approaches only partially address distribution mismatches between downstream tasks and CLIP's training data; the range of tasks they can handle is limited, and the hand-crafted rules they rely on are often inefficient. Therefore, we argue that it is necessary to design an approach that can effectively capture feedback signals from the retriever's training distribution and use them as supervision to guide query rewriting in a more reliable direction.

Inspired by Group Relative Policy Optimization (GRPO) (Shao et al., 2024), we address this challenge with GRAPE (Grouped Ranking-Aware Policy Optimization Enhancement), a plug-and-play and efficient enhancement approach that incorporates ranking signals into retrieval-guided query rewriting with LLMs. Intuitively, GRAPE proposes to leverage GRPO to bridge distributional differences—including language, length, and modality shifts—by transforming queries into forms better aligned with the retriever's training distribution. However, our preliminary experiment shows that directly finetuning LLM through similarity scores can lead to score inflation, where nearly all candidates are assigned unexpectedly high scores regardless of their true relevance. To address score inflation, we propose a corpus-relative ranking-based reward, which explicitly aligns optimization with ranking metrics while suppressing spurious score inflation. Extensive experiments demonstrate that GRAPE consistently improves retrieval performance under distributional shifts—including multilingual differences (Flickr30k-CN, CVLUE, XM3600), length differences (Wikipedia), and multimodal differences (CIRR)—achieving an average improvement of 4.9% in Recall@10. Our contributions are summarized as follows:

- We propose GRAPE, a plug-and-play retrieval enhancement approach that keeps the retriever frozen and improves retrieval through retrieval-guided query rewriting with LLMs, thereby avoiding costly re-embedding and redeployment.

- We introduce a corpus-relative ranking-based reward that explicitly aligns optimization with ranking objectives while effectively suppressing score inflation, a pitfall of similarity-based finetuning.

- We demonstrate that GRAPE achieves consistent and significant improvements across multiple distributional shifts—including multilingual, length, and multimodal differences—achieving an average 4.9% gain in Recall@10 on five representative benchmarks (Flickr30k-CN, CVLUE, XM3600, Wikipedia, CIRR) [1] .

## 2 RELATED WORKS

### 2.1 TRAINING-BASED RETRIEVAL

Training-based methods address the issue of input distributions diverging substantially from CLIP's training corpus by expanding the training data and subsequently finetuning the model. These methods generally fall into two categories: data-centric approaches, which focus on scaling or refining the training corpus, and model-centric approaches, which develop more sophisticated architectures to improve representation learning and cross-modal alignment.

**Data-centric approaches.** These methods focus on expanding or refining the training corpus. Large-scale efforts such as MetaCLIP-2 (Chuang et al.) and 100B-scale pre-training (Wang et al., 2025a) achieve notable gains in cross-lingual retrieval but demand massive resources. Other studies emphasize data quality, such as rewriting captions for richer supervision (Fan et al., 2023) or introducing composition-aware hard negatives to mitigate bag-of-words bias (Yuksekgonul et al., 2022). Although effective, these approaches are costly to construct and preprocess, and their improvements often remain confined to limited downstream tasks.

---

[1]The reproducible code and results are provided in the attachment and will be open-sourced after double-blind review.

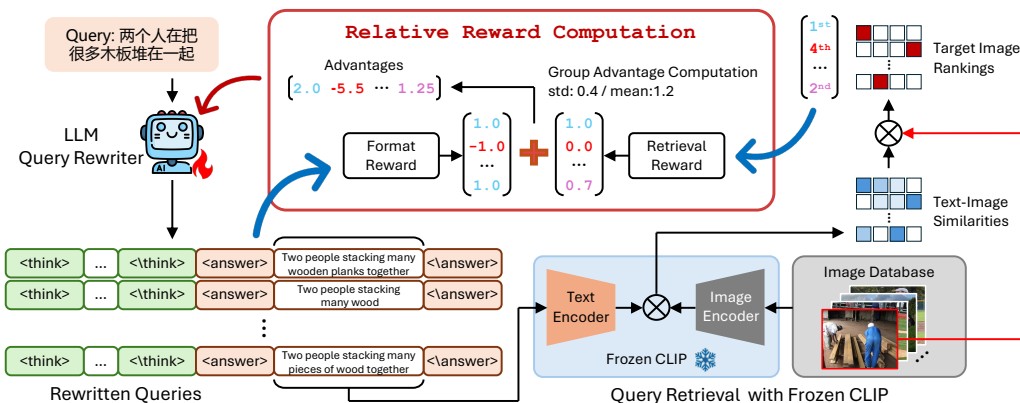

Figure 1: **Overview of GRAPE.** The framework operates in two phases: (i) *Query Rewriting and Retrieval*, where an LLM generates $K$ rewrites for a query, each encoded by a frozen CLIP retriever to produce target ranks; and (ii) *Relative Reward Computation and Optimization*, where a format reward and a rank reward are combined into a unified feedback signal, which is normalized within group to compute relative advantages and then used to update the LLM.

**Model-centric approaches.** These methods focus on optimizing internal representations or addressing structural constraints. Representative examples include BLIP-2 (Li et al., 2023), which introduces a trainable Q-Former to bridge frozen encoders with LLMs; finetuning strategies for pairwise reasoning (Sam et al., 2024) or compositional retrieval (Baldrati et al., 2023); and parameter-efficient tuning techniques such as adapters and LoRA (Wang et al., 2023). Other lines of work explore prompt learning (Zheng et al., 2025), long-text handling (Long-CLIP (Zhang et al., 2024), LoTLIP (Wu et al., 2024), FineLIP (Asokan et al., 2025)), multilingual adaptation (mCLIP (Chen et al., 2023), AltCLIP (Chen et al., 2022)), and multimodal fusion (CIR (Baldrati et al., 2023)). While these approaches achieve strong task-specific improvements, they inevitably incur retraining overhead and require re-generating large-scale pre-computed embedding databases.

## 2.2 TRAINING-FREE RETRIEVAL

Training-free methods aim to enhance CLIP's retrieval capability without modifying the encoder or regenerating large-scale embeddings, instead relying on external strategies that can be seamlessly integrated into existing systems. A representative direction focuses on query rewriting and augmentation. ComCLIP (Jiang et al., 2022) adopts a framework that decomposes both images and texts into semantic sub-parts and then aligns these components to achieve fine-grained alignment, but it is limited to queries and images whose semantics can be effectively decomposed. More recently, PlugIR (Lee et al., 2024) employs LLMs to iteratively refine user queries through dialogue and candidate context. However, owing to the lack of supervision, this approach relies on substantial hand-crafted constraints, which in turn limits its practicality in real-world deployments.

To overcome these limitations, we propose GRAPE, a plug-and-play enhancement approach that integrates ranking signals into retrieval-guided query rewriting with LLMs, enabling more effective and generalizable distribution adaptation without retriever retraining.

## 3 METHOD

In this section, we first introduce the problem formulation for enhancing retrieval systems (Section 3.1). We then present GRAPE (Section 3.2), a plug-and-play approach that employs GRPO to optimize query rewriting with ranking-based supervision. Finally, in Section 3.3, we explain the rationale for using ranking rather than similarity scores as the supervision signal in GRPO.

### 3.1 PROBLEM FORMULATION.

In CLIP-based retrieval tasks, given a query $q \sim \mathcal{D}_{\text{query}}$, the system embeds the query and each candidate item $i \in \mathcal{I} = \{i_1, \ldots, i_N\}$ into a shared semantic space and ranks candidates by similarity scores.

For text-to-image retrieval, let

$$z_q = f_{\text{text}}(q), \qquad z_i = f_{\text{img}}(i) \tag{1}$$

where $f(\cdot)$ denotes the CLIP encoder, $z_q$ and $z_i$ are $\ell_2$-normalized embeddings. The top-ranked item is selected by

$$i^\star = \arg\max_{i \in \mathcal{I}} s(z_q, z_i) \tag{2}$$

where $s(\cdot, \cdot)$ denotes the similarity function (e.g., cosine similarity). While CLIP achieves strong performance when queries follow its training distribution $\mathcal{D}_{\text{train}}$, it struggles under distributional shifts between the query distribution $\mathcal{D}_{\text{query}}$ and $\mathcal{D}_{\text{train}}$ (e.g., query length, multilingual inputs, or modality differences).

To address this issue, one approach is to expand $\mathcal{D}_{\text{train}}$ and finetune the retriever, but this incurs prohibitively high costs due to re-embedding and redeployment. An alternative is to leverage LLM knowledge to transform a query $q$ into a rewritten query $\tilde{q}$ that better matches $\mathcal{D}_{\text{train}}$. However, under the absence of supervision signals, such rewriting cannot fully align $\mathcal{D}_{\text{query}}$ with $\mathcal{D}_{\text{train}}$.

Therefore, our objective is to incorporate supervision signals into the distribution adaptation process, thereby enabling more effective query rewriting and bridge the distributional shifts.

### 3.2 GRAPE: GROUPED RANKING-AWARE POLICY OPTIMIZATION ENHANCEMENT

To address these challenges, we introduce GRAPE, a plug-and-play enhancement approach that incorporates ranking signals through retrieval-guided query rewriting with LLMs. Specifically, GRAPE rewrites a query $q$ into a group of queries $\tilde{q}_1, \ldots, \tilde{q}_k$ using an LLM, and leverages the relative ranking of the target image obtained from this group to update subsequent rewrites. The overall architecture is illustrated in Figure 1.

GRAPE operates in two phases: (i) Query Rewriting and Retrieval, where each query is rewritten into a group of queries and the frozen retriever yields the rankings of the target image; and (ii) Relative Reward Computation and Policy Optimization, where ranking results are used to compute relative rewards within the group and to optimize the rewriting LLM.

**Query Rewriting and Retrieval.** Given a raw query $q$, an LLM policy $\pi_\theta$ samples $K$ constrained rewrites $\{\tilde{q}_k\}_{k=1}^K$. After format validation, each rewrite is embedded by the *frozen* CLIP text encoder, $z_{\tilde{q}_k} = f_{\text{text}}(\tilde{q}_k)$; we then compute $s(z_{\tilde{q}_k}, z_i)_{i \in \mathcal{I}}$ against the precomputed candidate embeddings, rank all items accordingly, and obtain the target image rankings $\{r_k\}_{k=1}^K$.

**Relative Reward Computation and Optimization.** For every rewrite, GRAPE constructs two complementary reward. *Format reward:* Reasoning must be inside <think>...</think> and the final rewrite must be inside <answer>...</answer>. Non-conforming outputs receive $R^{\text{f}} = -1$ and are skipped (no CLIP retrieval); conforming outputs receive $R^{\text{f}} = 1$ and proceed to retrieval. *Ranking reward:* $R^{\text{r}}$ measures retrieval effectiveness using the rankings $\{r_k\}_{k=1}^K$. It is assigned after retrieval and increases monotonically with ranking quality (details in Section 3.3)

We integrate the two rewards into a unified feedback signal:

$$R_k = R_k^{\text{f}} + R_k^{\text{r}},$$

and compute group-wise statistics over the $K$ rewrites of the same query $q$:

$$\mu_q = \frac{1}{K} \sum_{k=1}^K R_k, \qquad \sigma_q^2 = \frac{1}{K} \sum_{k=1}^K \left(R_k - \mu_q\right)^2.$$

The relative advantage is

$$\tilde{A}_k = \frac{(R_k^{\text{f}} + R_k^{\text{r}}) - \mu_q}{\sqrt{\sigma_q^2}},$$

which emphasizes within-group improvements while stabilizing scale. The policy is updated with an advantage-weighted objective regularized toward a reference model $\pi_{\text{ref}}$:

$$
\mathcal{J}(\theta) = \mathbb{E}_{q \sim \mathcal{D}} \left[ \frac{1}{K} \sum_{k=1}^{K} \tilde{A}_k \log \pi_\theta(r_k \mid q, \mathcal{C}) \right] - \lambda \, \mathbb{E}_{q \sim \mathcal{D}} \left[ \text{KL}\big( \pi_\theta(\cdot \mid q, \mathcal{C}) \, \| \, \pi_{\text{ref}}(\cdot \mid q, \mathcal{C}) \big) \right], \quad (3)
$$

where only $\pi_\theta$ is updated; the CLIP encoders and corpus embeddings remain frozen.

### 3.3 Why Ranking Beats Score: Relative Ranking Reveals True Relevance

The ranking of the target image is the most direct indicator of retrieval quality, making it a natural signal for evaluating the effectiveness of LLM-based query rewrites. We therefore propose a ranking-based reward function to guide the rewriting process:

$$
R_k^{\text{r}} = 1 - \frac{2\,(r_k - 1)}{N - 1}, \quad (4)
$$

where $r_k \in \{1, \ldots, N\}$ denotes the ranking of the target among $N$ candidates.

As illustrated in Figure 1, for multiple rewrites $\{\tilde{q}_k\}_{k=1}^{K}$ generated from a single query $q$, those rewrites that better capture retrieval-relevant semantic details achieve higher ranking rewards. These true relevance encourage the model to update its rewriting policy toward improving retrieval performance. Moreover, Eq. 4 linearly maps rankings to rewards, directly reflecting changes in ranking through relative position. A detailed derivation is provided in the Appendix.

**Comparison with similarity-based rewards.** Similarity scores provide the most direct measure of the relationship between a query and the target image. However, a higher similarity score does not necessarily correspond to a higher ranking. For example, generic tokens such as "real-world" or "environment" can increase the similarity not only to the target image but also to many other real-world images. As a result, under the similarity-based reward

$$
R_k^{\text{s}} = s(z_{\tilde{q}_k}, z_t), \quad (5)
$$

the trained LLM often suffers from *score inflation*: nearly all candidates are assigned unexpectedly high scores regardless of their true relevance. In practice, the model can exploit this by injecting high-frequency but weakly discriminative tokens, which increase semantic relation without improving visual separability. Consequently, the model receives positive rewards, yet these signals do not guide it toward the correct optimization direction.

**Direct effect on optimization.** After within-group relative advantage computation, the effective driving signal is the *ranking difference* rather than the raw score gap. This guides the policy to generate rewrites that deliver genuine improvements in ranking order, rather than simply inflating similarity scores.

## 4 Experiments

In this section, we evaluate the effectiveness of our approach in addressing three major distributional differences commonly induced by queries in retrieval tasks: (1) **multilingual differences**—queries in languages that are underrepresented in CLIP's pretraining corpus; (2) **length differences**—query styles (e.g., long-form or complex phrasing) that deviate from the training distribution; and (3) **modality differences**—multimodal or compositional queries requiring reasoning beyond text-only inputs. Furthermore, to demonstrate the *data efficiency* of our method, we conduct experiments under varying proportions of training data. We also analyze the impact of similarity-based score inflation, verifying that our proposed ranking-based reward effectively suppresses this artifact.

### 4.1 Experimental Setup

**Datasets.** To evaluate the effectiveness of our method, we conduct experiments on five representative benchmarks, each designed to target one or more of the core challenges discussed above: multilingual differences, length differences, and modality differences.

- **Flickr30k-CN** (Lan et al., 2017): A Chinese extension of Flickr30k, designed for cross-lingual image–text retrieval.
- **CVLUE** (Wang et al., 2025b): A large-scale Chinese multilingual benchmark . It primarily addresses multilingual difference and Chinese culture understanding.
- **XM3600** (Thapliyal et al., 2022): A multilingual benchmark covering 36 languages, aimed at testing cross-lingual generalization and robustness.
- **Wikipedia** (Rasiwasia et al., 2010): An English text-to-image retrieval dataset featuring long and complex queries, making it suitable for evaluating length differences.
- **CIRR** (Liu et al., 2021): A fine-grained, compositional retrieval benchmark where both image and text serve as queries. It emphasizes modality discrepancies and requires nuanced semantic understanding. We follow an 80%/20% train/validation split for our experiments.

**Comparison Methods.**   We validate the scalability of our method across three CLIP variants: ViT-B/32, ViT-B/16, and ViT-L/14. For each backbone and dataset, we horizontally compare the performance of:

- **CLIP (baseline)**: frozen pretrained CLIP without query rewriting.
- **CLIP+LLM**: frozen CLIP with queries rewritten by a frozen LLM without retrieval feedback.
- **CLIP+GRPO-LLM**: frozen CLIP with queries rewritten by an LLM finetuned with GRPO, where training is conducted on the training set of the corresponding dataset using retrieval-based rewards.

**Evaluation Metrics.**   To achieve a more accurate evaluation, we calculate two key metrics respectively: the proportions of cases where the target image is included in the top-1 and top-10 retrieved results (i.e., R@1 and R@10).

**Implementation Details.**   Unless otherwise specified, we use `Qwen2.5-3B-Instruct` (Yang et al., 2024) as the query–rewriting model across the four datasets. For CIRR that requires multimodal query understanding, we adopt the vision–multilingual model `Qwen2-VL-7B` (Wang et al., 2024) as the rewriter. We employ three task-specific prompt templates, and we keep the prompts identical for training and validation. More details are presented in the Appendix.

## 4.2   MAIN RESULTS

Table 1: R@1 and R@10 for different CLIP model scales and methods. "+LLM" indicates frozen LLM rewriting, while "+GRAPE" denotes GRPO-finetuned rewriting (ours). GRAPE consistently achieves notable improvements over both vanilla CLIP and CLIP+LLM across all benchmarks and model sizes, delivering significant gains in R@1 and R@10. A dash ("–") indicates that the corresponding model is not capable of handling the task directly.

| Model | Flickr30k-CN T2I | | CVLUE T2I | | XM3600 T2I | | Wikipedia T2I | | CIRR TI2I | | Average | |
|---|---|---|---|---|---|---|---|---|---|---|---|---|
| | R@1 | R@10 | R@1 | R@10 | R@1 | R@10 | R@1 | R@10 | R@1 | R@10 | R@1 | R@10 |
| ViT-B/32 | — | — | — | — | 11.4 | 25.5 | 28.4 | 65.8 | — | — | 19.9 | 45.6 |
| +LLM | 49.8 | 84.4 | 10.6 | 38.5 | 49.5 | 77.8 | 30.4 | 67.1 | 29.5 | 70.5 | 34.0 | 67.7 |
| **+GRAPE(↑)** | **53.3** | **87.3** | **13.1** | **44.4** | **56.0** | **83.1** | **36.1** | **78.4** | **34.3** | **77.8** | **38.6** | **74.2** |
| ViT-B/16 | — | — | — | — | 12.3 | 26.8 | 33.5 | 71.9 | — | — | 22.9 | 49.4 |
| +LLM | 52.1 | 86.0 | 12.1 | 42.0 | 52.9 | 78.9 | 34.3 | 69.3 | 30.7 | 73.3 | 36.4 | 69.9 |
| **+GRAPE(↑)** | **58.0** | **90.0** | **14.6** | **48.7** | **58.4** | **83.1** | **43.7** | **82.8** | **35.7** | **77.8** | **42.1** | **76.5** |
| ViT-L/14 | — | — | — | — | 14.2 | 29.0 | 40.4 | 80.2 | — | — | 27.3 | 54.6 |
| +LLM | 57.6 | 89.5 | 13.7 | 43.6 | 51.7 | 78.1 | 42.0 | 77.6 | 29.7 | 70.1 | 38.9 | 71.9 |
| **+GRAPE(↑)** | **62.6** | **92.6** | **15.3** | **48.9** | **58.1** | **82.6** | **47.8** | **83.5** | **33.2** | **76.3** | **43.4** | **76.8** |

As shown in Table 1, the distribution gap between training data and queries limits CLIP's ability to handle cross-lingual and multimodal inputs, and even for long-text queries the recall rate remains

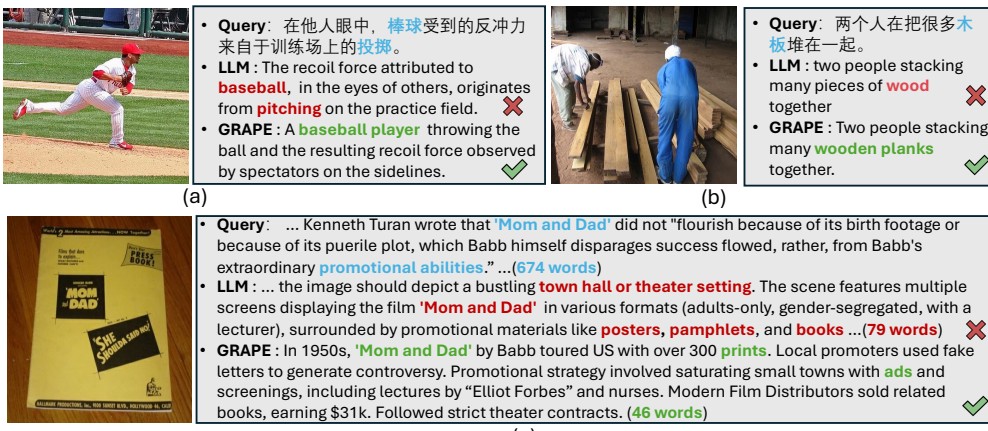

(a)                                    (b)

(c)

Figure 2: Case study of query rewriting, where Figures (a) and (b) illustrate multilingual query cases, and Figure (c) shows a case with a long-form query. The key concepts in the original query are marked blue; literal or incomplete rewrites generated by a vanilla LLM are marked red; retrieval-friendly expressions produced by GRAPE are marked green.

relatively low. While query rewriting with LLMs can partially mitigate this gap, the absence of optimization guidance often leads to suboptimal rewrites. In contrast, our proposed GRAPE framework effectively aligns query distributions with the retriever, yielding consistent improvements across all settings. On average, GRAPE achieves gains of more than 4.5% in R@1 and 4.9% in R@10 over CLIP+LLM, and these improvements hold across different CLIP variants, with stronger models achieving higher absolute recall under GRAPE enhancement.

### 4.3 CRITICAL ANALYSIS

Why can GRAPE address the three major distributional differences commonly induced by queries in retrieval tasks? To answer this, we conduct analytical experiments on how GRAPE overcomes the challenges that arise when adapting query distributions: multilingual differences, by performing cross-lingual translation and semantic enrichment; length differences, by conducting long-text distillation and expansion; and modality differences, by enabling cross-modal understanding and generating retrieval-friendly expressions.

**Cross-Lingual Query Translation and Semantic Enrichment.** This challenge arises not only from the difficulty of accurate translation, but also because direct translations often overlook implicit semantics required for retrieval, resulting in literal yet incomplete queries. As illustrated in Fig. 2(a), the original query implicitly refers to a "baseball player," yet a vanilla LLM produces only a literal translation that misses this hidden entity. Similarly, in Fig. 2(b), "wooden planks" is simplified to "wood," discarding crucial descriptive details. In contrast, GRAPE generates a retrieval-oriented rewrite that explicitly recover latent entities and enrich semantic details (e.g., "baseball player throwing a ball," "wooden planks"), thereby strengthening cross-lingual rewrite.

**Long-Text Expression Distillation and Expansion.** This challenge arises because long-text queries often contain excessive redundancy that buries the salient concepts, making it difficult for CLIP to capture the true retrieval intent. At the same time, such inputs may also lack explicit emphasis on key entities, leaving gaps in semantic coverage. As shown in Fig. 2(c), a 674-word Wikipedia passage about the film *Mom and Dad* leads a vanilla LLM to generate a verbose 79-word rewrite that even hallucinates additional screening details. In contrast, GRAPE performs both distillation, by removing redundant descriptions, and enrichment, by expanding essential concepts (e.g., *Mom and Dad*, *prints*, *ads*), thus producing concise yet sufficiently informative queries. This synergy between distillation and enrichment enhances retrieval performance on long-text benchmarks.

**Multimodal Understanding and Efficient Expression.** This challenge arises because multi-modal queries—which CLIP alone struggles to handle—require the integration of both image and

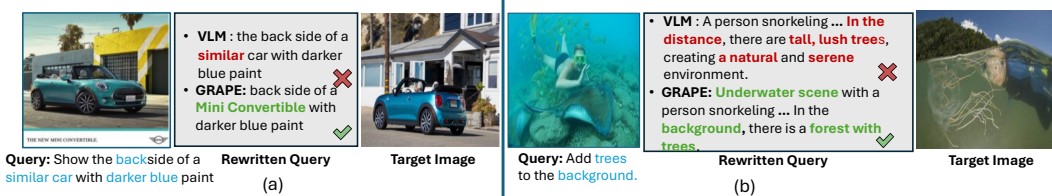

Figure 3: Case study of multimodal query rewriting. Since CLIP does not natively support multimodal inputs, image information must first be extracted and then integrated into the text according to the instruction.

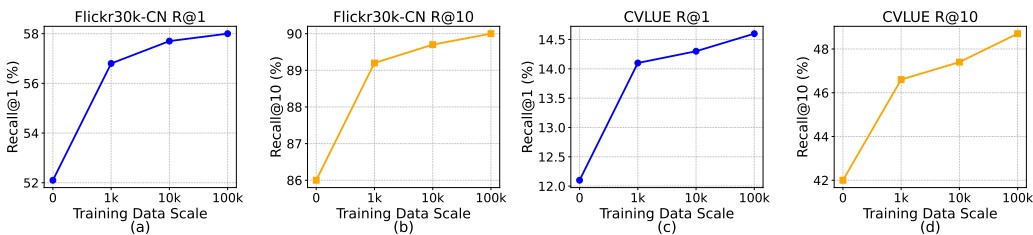

Figure 4: R@1 and R@10 performance across different training data scales for Flickr30k-CN and CVLUE. GRAPE demonstrates data efficiency, where performance with fewer training samples approaches that of larger datasets, and consistently outperforms the untrained LLM.

text information. Combining visual and textual inputs demands accurate cross-modal understanding and the generation of efficient representations. However, vanilla VLMs often fail to capture the key cross-modal signals and thus struggle to produce retrieval-friendly expressions. As shown in Fig. 3(a), a vanilla VLM cannot correctly resolve that the textual phrase "similar car" refers to the "Mini Convertible" in the image. As shown in Fig. 3(b), though fusion occurs, the generated description of "tree" is overly complicated and deviates from the retrieval intent. In contrast, GRAPE learns to effectively integrate multimodal information into unified, retrieval-friendly queries, capturing complementary semantics and thereby enhancing retrieval performance in multimodal scenarios.

## 4.4 ABLATION STUDY

**Data Efficiency of GRAPE.** GRAPE supervises the LLM to rewrite queries using ranking signals. Unlike conventional supervised finetuning, which often requires the LLM to acquire additional domain knowledge, GRAPE only requires the model to learn how to generate retrieval-friendly rewrites for the target retrieval task. To validate the data efficiency of GRAPE, we conduct experiments on Flickr30k-CN and CVLUE using the ViT-B/16 backbone with training subsets of different sizes. As shown in Fig. 4, GRAPE consistently outperforms pretrained LLMs (with zero additional training data) across all data scales. Specifically, on Flickr30k-CN (Fig. 4(a)), GRAPE achieves R@1 that reaches 94% of the full 100k-sample performance with only 10k samples. Similarly, on CVLUE (Fig. 4(c)), GRAPE achieves 88% of the full-data performance with 10k samples.

**Limitations of Similarity-Based Supervision.** Directly finetuning LLMs with similarity scores often leads to score inflation. This issue is primarily caused by sample-wise isolated measurement, which overlooks the influence of other images in the retrieval corpus. To validate this issue, we conducted experiments on Flickr30k-CN using ViT-B/16 as the backbone, where similarity scores defined in Eq. 5 were used as the reward. As shown in Fig. 5, the average similarity between rewritten queries and target images steadily increases with training steps, while R@1 decreases correspondingly. In practice, the model tends to output generic words such as "environment" or "image," which increase similarity with the target image but simultaneously with many irrelevant images. A detailed comparison between similarity-based and ranking-based rewards is provided in Appendix A.4.

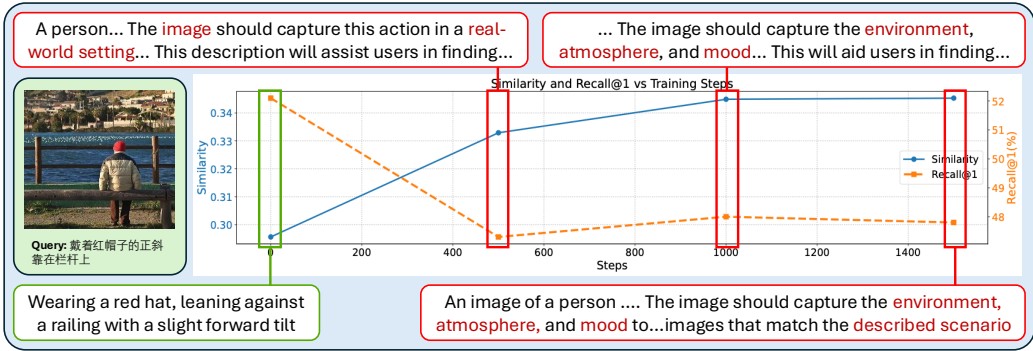

Figure 5: Using similarity scores as the reward leads to score inflation. As training epochs increase, the similarity between the query and the target image improves, but Recall@1 steadily decreases. The boxed text shown in the figure corresponds to the rewritten queries generated at the indicated training steps. Due to the lack of supervision signals, the LLM tends to generate generic or irrelevant words that raise similarity scores without improving retrieval quality.

## 5 DISCUSSION

**Coverage of Downstream Tasks.** GRAPE enhances retrieval by relying on the LLM's knowledge to rewrite queries into forms closer to CLIP's training distribution. This design inherently limits its effectiveness to the knowledge capacity of the LLM. When downstream tasks require knowledge beyond the LLM's coverage, the model may fail to produce meaningful rewrites, ultimately constraining GRAPE's effectiveness. For example, in multilingual retrieval, the LLM exhibits weaker capability for low-resource languages, which in turn restricts the benefits of GRAPE; detailed results are provided in the Appendix.

**Upper Bound by the Retriever.** GRAPE leverages ranking signals from a frozen retriever as optimization feedback. This design enables our method to adapt queries without modifying the embedding space, but it also implies that the performance ceiling is constrained by the capacity of the underlying retriever. As a result, if CLIP lacks sufficient expressiveness for certain tasks, GRAPE can help close the gap but cannot surpass the retriever's intrinsic upper bound. Nevertheless, as the representational power of CLIP improves, GRAPE shows consistent gains in performance.

**Time and Efficiency Considerations.** Since all query are rewritten by the LLM before retrieval, our approach inevitably introduces additional inference latency. While the extra cost is modest in controlled benchmarks, scaling to large-scale industrial retrieval systems raises practical concerns. Balancing retrieval accuracy gains with computational overhead and latency constraints remains an important direction for future work.

## 6 CONCLUSION

In this work, we address the critical challenge of enhancing CLIP-based retrieval systems under distributional shifts without costly re-embedding or redeployment. We introduce **GRAPE**, a plug-and-play approach that leverages LLM-based query rewriting guided by retrieval feedback. By incorporating a corpus-relative ranking-based reward, GRAPE explicitly aligns optimization with ranking objectives and, importantly, avoids the pitfall of similarity-based supervision—*score inflation*, where nearly all candidates receive high scores regardless of true relevance. Extensive experiments across five representative benchmarks demonstrate that GRAPE consistently improves retrieval performance under multilingual, length, and multimodal shifts, yielding an average gain of 4.9% in Recall@10. These results highlight the effectiveness of ranking-aware supervision in bridging distributional gaps while preserving compatibility with frozen retrievers. We believe our findings provide a promising direction for building adaptable retrieval enhancement methods at scale, without requiring retriever retraining or re-embedding.

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

# A APPENDIX

## A.1 USE OF LLMS

Large language models (LLMs) are used as query rewriters in our experiments. They are also used to assist in manuscript preparation, such as polishing the text and suggesting references during the literature review. No part of the core methodology, reward design, evaluation protocol, or experimental results is generated or influenced by LLMs.

## A.2 ETHICS STATEMENT

Our study investigates retrieval enhancement under distributional shifts by leveraging supervised query rewriting guided by ranking signals. No human subject or private data is involved. All datasets used are from publicly available benchmarks, and all evaluated models (CLIP and LLM backbones) are publicly released. While our analysis reveals potential weaknesses of existing retrieval systems, our intent is solely to advance research toward more reliable and equitable retrieval methods, rather than to exploit such weaknesses.

## A.3 REPRODUCIBILITY STATEMENT

We provide code and data to reproduce all major results. Our supplementary materials include: (1) code for training and evaluation with ranking-based rewards, (2) the full prompt templates used in different tasks, and (3) detailed training configurations and dataset splits. Complete implementation details and extended results will be released upon acceptance.

## A.4 ANALYSIS OF THE RANKING-BASED REWARD FUNCTION

We prove that the relative advantage computation in GRPO is affine-invariant with respect to the reward scale. Even if rankings are linearly mapped to scores, the final normalized signal depends only on relative ordering.

**Ranking-to-Reward Mapping.** Given a ranking $r_k \in \{1, \ldots, N\}$, consider a general affine mapping:
$$R_k = ar_k + b, \quad a \neq 0.$$
Our specific reward $R_k^r = 1 - \frac{2(r_k-1)}{N-1}$ is a special case.

**Relative Advantage Normalization.** For a query $q$ with $K$ rewrites $\{R_k\}_{k=1}^K$, GRPO computes
$$\tilde{A}_k = \frac{R_k - \mu_q}{\sigma_q}, \quad \mu_q = \frac{1}{K}\sum_{j=1}^K R_j, \quad \sigma_q^2 = \frac{1}{K}\sum_{j=1}^K (R_j - \mu_q)^2.$$

**Affine Invariance.** Substituting $R_k = ar_k + b$ gives
$$\mu_q = a\mu_r + b, \qquad \sigma_q = |a|\sigma_r,$$
where $\mu_r$ and $\sigma_r$ are the mean and std of $\{r_j\}$. Thus,
$$\tilde{A}_k = \frac{a(r_k - \mu_r)}{|a|\sigma_r} = \text{sign}(a) \cdot \frac{r_k - \mu_r}{\sigma_r}.$$

**Conclusion.** The normalized advantage $\tilde{A}_k$ is invariant under affine transformations of $R_k$, up to a global sign flip. Hence, the optimization depends only on ranking order, ensuring robustness of our ranking-based reward formulation.

**Empirical Comparison with Similarity-based Rewards.** To complement the above proof, we conduct experiments comparing ranking-based and similarity-based rewards as training progresses. Figure 6 shows that similarity-based rewards exhibit score inflation (average similarity increases but Recall@1 decreases), ranking-based rewards yield stable improvements in retrieval recall with training steps. This demonstrates that our ranking-based reward not only has desirable theoretical properties but also provides practical stability during optimization.

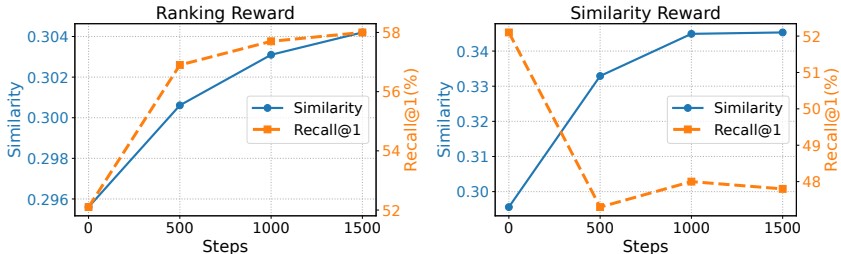

Figure 6: Comparison of similarity-based and ranking-based rewards during training. Ranking-based rewards avoid score inflation and align with improvements in retrieval recall.

## A.5 EXPERIMENTAL SETUP DETAILS

**Datasets.** Table 2 summarizes the training and validation splits of all datasets used in our experiments, including the number of images and queries for each split.

Table 2: Training and validation statistics of the datasets used in our experiments.

| Dataset | Train Queries | Train Images | Val Queries | Val Images |
|---|---|---|---|---|
| Flickr30k-CN | 148,915 | 29,783 | 5,000 | 1,000 |
| CVLUE | 89,600 | 17,920 | 15,580 | 3,116 |
| Wikipedia | 2,173 | 2,173 | 693 | 693 |
| XM3600 | 126,341 | 1,745 | 33,218 | 456 |
| CIRR | 3,336 | 1,951 | 834 | 723 |

**Training Setup.** GRAPE is trained on each dataset using 1.5k or 2.5k optimization steps, where training with smaller datasets typically converges within 1.5k steps and training with larger ones requires up to 2.5k steps. We adopt a batch size of 8 with gradient accumulation, and optimize the model using AdamW with a learning rate of $5 \times 10^{-7}$ under a cosine schedule and a warmup ratio of 0.03. The frozen retrievers include ViT-B/32, ViT-B/16, and ViT-L/14. To ensure fairness, the same prompt templates are used consistently across both training and validation.

**Prompt Templates.** We designed task-specific prompts for different scenarios: multilingual retrieval, multimodal fusion, and long-form text queries. These templates guide the LLM rewriter to generate retrieval-friendly inputs for CLIP.

## A.6 EXTENDED EXPERIMENTAL RESULTS AND ANALYSIS

**Effect of Knowledge Coverage.** We provide per-language retrieval performance on XM3600 using radar plots in Figure 7, reporting Recall@1 and Recall@10 across 36 languages. The plots highlight variation between high-resource and low-resource languages: GRAPE yields strong improvements when the LLM has sufficient knowledge coverage (e.g., English, Chinese, Spanish), but the gains diminish for underrepresented languages. This further demonstrates that the coverage of downstream tasks by GRAPE is ultimately bounded by the knowledge reserves of the LLM. Nevertheless, as LLM capabilities continue to improve, the range of downstream tasks that GRAPE can effectively cover is expected to expand accordingly.

Table 3: Prompt templates used for different retrieval scenarios.

| Task | Prompt Template |
|------|-----------------|
| Multilingual | `You're an image retrieval assistant.  Translate search queries:  {text} into optimized English text for vector-based image search.  Show your work in <think></think> tags.  And return the final text in <answer></answer> tags.` |
| Multimodal | `You're an image retrieval assistant.  You need to use the information in the image, generate a text based on the description in the text: {text}.  This generated text will be used by CLIP to retrieve the corresponding image.  Show your work in <think></think> tags.  And return the final text in <answer></answer> tags.` |
| Length | `You are an image retrieval assistant.  Summarize the given Wikipedia text content:  {text} into a concise visual description suitable for CLIP model input to retrieve corresponding images of the described subject.  Show your work in <think></think> tags.  And return the final text in <answer></answer> tags.` |

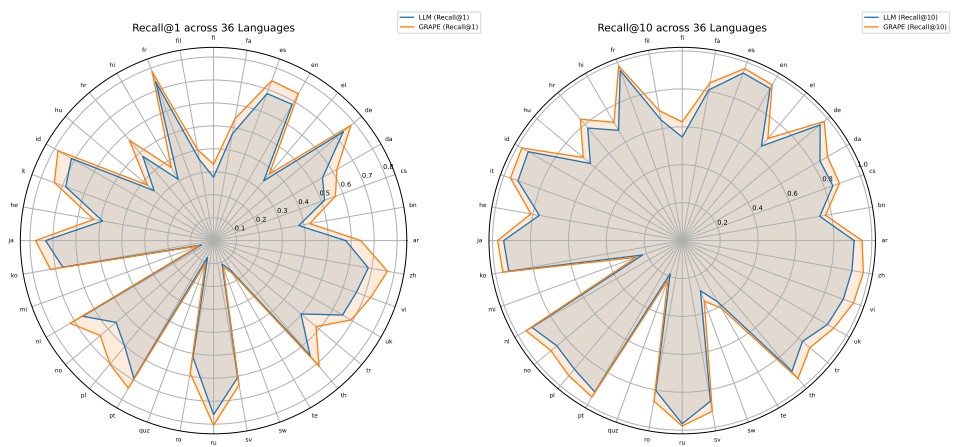

Figure 7: Radar plots of Recall@1 (top) and Recall@10 (bottom) for GRAPE on XM3600 across 36 languages. Performance is stronger in high-resource languages and weaker in low-resource ones, reflecting the limitations imposed by LLM knowledge coverage.

