# OpenReview forum: "GRAPE: Let GPRO Supervise Query Rewriting by Ranking for Retrieval"
_ICLR.cc/2026/Conference — ICLR 2026 Conference Withdrawn Submission_

### Official Review · Reviewer_8h3a · 2025-10-23

**Soundness:** 3
**Presentation:** 3
**Contribution:** 2
**Rating:** 4
**Confidence:** 4

**Summary:**

**Motivation**: CLIP retrievers work well in-distribution but degrade when queries are longer, multilingual, or multimodal. Retraining the retriever to cover these shifts is costly because it needs re-embedding and redeploying large indexes, and naive LLM-based query rewriting lacks supervision to reliably fix the gap. The paper aims to add a simple supervision signal to query rewriting so the rewritten queries better match the retriever’s training distribution.

**Approach**: GRAPE uses an LLM to produce K constrained rewrites for each query, scores each rewrite with a frozen CLIP retriever, and then learns from two rewards: a format reward that checks output structure and a ranking reward that increases when the target ranks higher. It aggregates rewards within the group of K rewrites, normalizes them, and updates the LLM with a GRPO objective regularized toward a reference model, keeping CLIP encoders and corpus embeddings frozen. The method is plug-and-play at the query layer and trains only the rewriting policy.

**Key Results**: Across four benchmarks that stress different shifts—multilingual (CVLUE, XM3600), long queries (Wikipedia), and multimodal queries (CIRR)—GRAPE raises R@1 and R@10 over both vanilla CLIP and frozen LLM rewriting, for all tested CLIP backbones. The paper reports consistent gains in the main table and positions the method as an effective, retriever-frozen upgrade path.

**Strengths:**

**Deployment-friendly and low risk:** GRAPE keeps the CLIP encoder and all index embeddings frozen, so you do not need to re-index or redeploy the corpus. It works as a drop-in layer at query time by rewriting the query and reusing the existing retriever. The setup is compatible with different CLIP backbones and is easy to roll back if needed.

**Clear, stable learning signal:** Training uses group-wise GRPO with a rank-based reward tied to the actual retrieval order, plus a simple format check so outputs stay well formed. Rewards are normalized within each group of K rewrites, which helps stabilize updates and prevents reward scale drift.

**Consistent gains and good data efficiency, with analysis that explains why:** Across multilingual, long-text, and multimodal settings, GRAPE shows steady R@1 and R@10 improvements for multiple model sizes, along with case studies that match the aggregate results. Learning curves indicate most of the benefit arrives with modest training data, which makes the method practical. The paper also explains a common failure mode of similarity-based rewards that inflate scores yet hurt recall, and shows why a ranking-based objective avoids this problem.

**Weaknesses:**

**Lack of novelty and missing baselines:** Several prior works already target query rewriting to improve retrieval and RAG, including trainable rewriters and zero-shot reformulation methods. Given this history, the paper’s contribution feels incremental unless the GRPO with explicit ranking reward is shown to be the key driver. To validate novelty, the authors should compare against stronger rewriting baselines beyond a frozen LLM rewrite, for example HyDE-style reformulation, multi-query rewriting, RL-trained rewriters, and recent RQ-RAG variants, under matched token budgets and latency. Ablations should isolate the value of ranking vs similarity rewards, group normalization, and the number of rewrites K across multiple retrievers.

**Latency**: Every query is rewritten K times, so both token cost and wall-clock time grow roughly linearly with K. Each rewrite also triggers its own CLIP scoring and a merge of K result lists, which can raise latency. It would help to evaluate small rewriters, dynamic K, early stopping for weak rewrites, and caching to reduce redundant computes, and to include a cost–quality curve so readers can pick an operating point.

**Reward not tailored to multimodal and narrow evaluation scope:** The method trains the rewriter with a rank-based reward that is modality-agnostic; it does not use signals that are specific to cross-modal alignment or image semantics. Given that, it is unclear why the experiments are limited to CLIP-only setups, rather than also testing text-only or code retrievers to show generality. If the goal is CLIP adaptation, a multimodal-aware objective or auxiliary checks grounded in the image space would make the design feel more aligned with the task. If the goal is a general rewriting recipe, the paper should include non-CLIP retrievers to demonstrate that the approach is not tied to a single modality.

**Questions:**

1. How should one pick K (number of rewrites) and group size to balance recall gains vs latency for real systems?
2. How well does GRAPE transfer to other retrievers and tasks, like text-only retrieval, code search, or very long document search without image targets?

---

### Official Review · Reviewer_KhMa · 2025-10-29

**Soundness:** 2
**Presentation:** 2
**Contribution:** 2
**Rating:** 4
**Confidence:** 3

**Summary:**

In this paper, the authors proposed a method called GRAPE, which aims to enhance CLIP-based retrieval under distribution shifts—such as multilingual, long-text, or multimodal queries—by training an LLM to rewrite queries using ranking feedback from a frozen retriever, thereby improving alignment with CLIP’s training distribution without modifying or re-embedding the database. By leveraging group-relative ranking rewards, GRAPE avoids similarity-score inflation and consistently boosts recall across multiple benchmarks, with only modest additional inference overhead.

**Strengths:**

+ This paper identifies a problem in retrieval systems known as score inflation, revealing the fragility of supervision signals in current retrieval model fine-tuning methods that predominantly rely on similarity scoring.
+ To address this issue, the paper proposes a solution called GRAPE, which replaces similarity scores with retrieval ranking to achieve retrieval-based supervision.

**Weaknesses:**

+ The experimental design in this paper suffers from serious flaws. The extremely limited number of baselines undermines the data-supported analysis throughout the paper, and conducting experiments solely on the QWEN model significantly weakens the persuasiveness of the findings. It is particularly unreasonable to include CLIP directly as one of the baselines, given that the study focuses on the impact of using similarity scores as supervision signals and the effectiveness of the newly proposed ranking-based supervision method for retrieval. The baselines should have been selected from methods that also utilize supervision signals, but the chosen baselines do not adequately address the experimental requirements.
+ Furthermore, many conditions across the baselines should have been kept consistent, yet the CLIP baseline does not even employ query rewriting, making its inclusion questionable.
+ Additionally, the ablation experiments in this paper are designed from an overly narrow perspective. Aspects such as data efficiency and the reasons behind the poor performance of similarity-based scoring do not constitute the core focus of ablation studies—the latter can hardly even be considered an ablation experiment. Ablation experiments should demonstrate the necessity and contribution of each component within the proposed framework by systematically removing or altering parts of it and evaluating the overall performance, thereby proving that the integrated design achieves optimal results. However, the proposed framework does not include similarity-based scoring as a component, indicating a misunderstanding of the purpose of an ablation study.

**Questions:**

See above

---

### Official Review · Reviewer_auro · 2025-10-31

**Soundness:** 3
**Presentation:** 4
**Contribution:** 3
**Rating:** 4
**Confidence:** 3

**Summary:**

This paper presents GRAPE, a query-rewriting approach designed to enhance CLIP-based retrieval on domains that lie outside CLIP’s original training distribution. Instead of retraining the CLIP model, GRAPE trains a language model rewriter using GRPO and a ranking-based reward that encourages rewritten queries to yield better retrieval rankings. Experiments across multiple datasets demonstrate consistent improvements, showing that GRAPE effectively adapts CLIP to new domains without the need for retraining or re-indexing.

**Strengths:**

* The paper is very well written, the flow is clear, the motivation well explained.
* The results are quite impressive, with a large set of evaluation tasks covering multiple cases and multiple CLIP models.
* The method enables quick adaptation of CLIP to other domains.

**Weaknesses:**

* Overall, this paper can be seen as just "GRPO with a retrieval signal as reward" thus the methodological contribution might seem limited (although it works well, so it's not critical imo).
* Baselines feel a bit limited: I would have preferred to see more baselines, e.g.:
   * More prompts. Your experiments using prompting alone—without any training—yield relatively low results. How much room for improvement do you think there is by refining the prompt itself, perhaps even automating this process by asking a model like GPT to generate optimized prompts from a few examples?
   * An obvious baseline is to fine-tune CLIP itself and compare. I agree it requires indexing of the collection (which, although offline, is expensive) but it would provide an interesting comparison point (likely an upper bound).
   * Larger LLMs: In particular, I think we would need to see zero-shot results with a SoTA LLM and an improved prompt, just for comparison.
* The claim that similarity as a reward is not a good choice is true but not so useful. Of course: if all documents/queries share the same embedding then this criterion is perfectly optimized…
* One of the strong claim is that this method prevents re-indexing. However this points is a bit ambiguous. If applying the CLIP model on a new domain, one could argue that no index already exists and this not an actual problem. On the contrary, if for some existing index there exists a subtask which is not well addressed, then it becomes ambiguous as to when using the query rewriting? In particular, if using it on data which was previously well handled. In clear, I’m wondering about how practical this is and if there is a real case scenario where an existing index can be realistically kept.
* Experiments train task-specific GRAPE models. In real life applications, it may very well be that we want to keep an index but have multiple new incoming tasks on which we want to improve. I have doubts regarding the performances of GRAPE in such a case.

**Questions:**

Additional questions:

* You show that using the similarity of the positive document as reward signal leads to similarity inflation. What about using a contrastive reward signal, i.e., for a batch of data, much like CLIP, measure the constrastive loss? This way, the loss signal would include incentive to deflate some of the similarities (the ones of the negative images selected in the batch). It seems to me that the similarity inflation you observe can likely come from a mode collapse: the ensuing reward is maximal if all documents/queries have the same embedding.
* Could you provide results when fine-tuning the CLIP model on the training data of each dataset (when it’s possible)?
* For each new data you obtain a query-rewriting model. Is there some generalization properties of these models? CLIP is very general-purpose, while your adaptation of it is very domain-specific: is there an in-between here? It seems there could be, as being "retrievally descriptive" of some input is a fairly generic task (e.g., humans do it well).
* Is the reasoning part of the template useful? Or did you use it only because it was a part of Qwen3?
* Reproducibility is probably a bit difficult, being GRPO and all… For completeness, could you provide results with a different backbone (non-Qwen) on at least one of the datasets?
* Your method may be a bit more generic than just a CLIP add-on. Do you think you could show applications of this to other retrieval models (e.g., convert a monolingual text retriever into a multilingual one?)
* Regarding the re-indexing: is it absolutely necessary when fine-tuning CLIP? How bad is the performance degradation on a held-out task if you fine-tune on another task? Same question if you fine-tune while keeping also part of the training data in the training mix? It would be an "ugly" solution, but if it works it remains the simplest.

---

### Official Review · Reviewer_Dj4E · 2025-11-01

**Soundness:** 2
**Presentation:** 2
**Contribution:** 2
**Rating:** 2
**Confidence:** 3

**Summary:**

The paper introduces GRAPE (Grouped Ranking-Aware Policy Optimization Enhancement), a plug-and-play method to improve CLIP-based retrieval systems without retraining the retriever or indexer. GRAPE fine-tunes a query-rewriting LLM via Group Relative Policy Optimization (GRPO). For each query, K rewritten queries are generated, and group-wise normalization is applied to compute relative advantages for policy updates.

**Strengths:**

1. The paper proposes GRAPE, which targets distribution shift in deployed CLIP systems and avoids re-training costs of retriever;
2. The input queries are rewritten according to one format reward and one rank reward;
3. GRAPE improves R@1/R@10 on multiple datasets;
4. Qualitative analyses are provided. Case studies demonstrate how GRAPE enhances query understanding by adding missing semantics and integrating visual context into textual descriptions.

**Weaknesses:**

1. The paper does not provide a detailed analysis of computational complexity (e.g., parameter count, training/inference time, or memory footprint)
2. The evaluation lacks comparison with strong or recent state-of-the-art baselines (see Table 1), which weakens the empirical justification
3. The ranking reward in Eq. (4) is purely linear, failing to emphasize top-ranked transitions (e.g., Top-1 vs. Top-2) or generalize to multi-positive retrieval scenarios. Moreover, the reward scale depends on the number of candidates $N$, with no normalization or robustness analysis
4. The paper linearly combines the ranking-based reward $R^r$ and the format reward $R^f$ without analyzing their relative weighting, interaction, or complementary roles. The motivation and theoretical grounding of these two components are insufficiently justified.

**Questions:**

1. Could the authors provide an analysis of the model’s parameter size, training/inference time, and memory footprint to better assess its efficiency compared to existing methods?
2. What is the rationale for linearly combining the ranking-based reward $R^r$ and format reward $R^f$? Have the authors analyzed their relative contributions or considered alternative weighting or normalization strategies?

---

### Note · Authors · 2025-11-13

I have read and agree with the venue's withdrawal policy on behalf of myself and my co-authors.